# Sustainability of a Compartmentalized Host-Parasite Replicator System under Periodic Washout-Mixing Cycles

**DOI:** 10.3390/life8010003

**Published:** 2018-01-26

**Authors:** Taro Furubayashi, Norikazu Ichihashi

**Affiliations:** 1Graduate School of Frontier Biosciences, Osaka University, 1-5 Yamadaoka, Suita, Osaka 565-0871, Japan; fbayashi@fbs.osaka-u.ac.jp; 2Department of Bioinformatics Engineering, Graduate School of Information Science and Technology, Osaka University, 1-5 Yamadaoka, Suita, Osaka 565-0871, Japan

**Keywords:** parasites, compartments, sustainability, origin of life, primitive replicators

## Abstract

The emergence and dominance of parasitic replicators are among the major hurdles for the proliferation of primitive replicators. Compartmentalization of replicators is proposed to relieve the parasite dominance; however, it remains unclear under what conditions simple compartmentalization uncoupled with internal reaction secures the long-term survival of a population of primitive replicators against incessant parasite emergence. Here, we investigate the sustainability of a compartmentalized host-parasite replicator (CHPR) system undergoing periodic washout-mixing cycles, by constructing a mathematical model and performing extensive simulations. We describe sustainable landscapes of the CHPR system in the parameter space and elucidate the mechanism of phase transitions between sustainable and extinct regions. Our findings revealed that a large population size of compartments, a high mixing intensity, and a modest amount of nutrients are important factors for the robust survival of replicators. We also found two distinctive sustainable phases with different mixing intensities. These results suggest that a population of simple host–parasite replicators assumed before the origin of life can be sustained by a simple compartmentalization with periodic washout-mixing processes.

## 1. Introduction

The primitive replicators during the origin of life [1,2,3,4,5,6,7] are supposed to have suffered from exploitation by parasitic replicators [8,9,10]. Parasitic replicators inevitably emerge from catalytically active replicators (or “host” replicators) by deleterious mutations and use up replication resources, such as replication enzymes and nutrients. As a result, parasitic replicators dominate the system and strongly inhibit the replication of host replicators, often leading to the eventual extinction of all the replicators. A plausible strategy for primitive host replicators to avoid parasite dominance is compartmentalization [11,12,13,14,15,16,17,18,19] or equivalent spatial structures [16,20]. Compartmentalization might have been achieved in the primordial earth by formation of lipid vesicles [21,22], coacervates [23,24], inorganic compartments at hydrothermal sites [25], honeycomb structure on mineral surface [26], atmospheric aerosols [27], hydrogels [28], or any other abiotic compartments provided from the surrounding environment. 

Although it is experimentally confirmed that compartmentalization slows down parasite dominance [29], a compartmentalized host-parasite replicator (CHPR) system needs to purge emerged parasites for long-term survival, a prerequisite for life. A simple way to purge parasites is periodic washout and mixing of a population of compartments, which might have been caused by periodic environmental fluctuations, such as high-low tidal cycles. Indeed, recent experimental studies [30,31], where a population of micro-droplets containing RNA templates and Qβ replicases undergoing cycles of washout and mixing, showed that the washout-mixing cycles prevented parasite dominance-induced immediate extinction of all the replicators. However, due to the limitations of time and conditions that experiments can mimic, the long-term sustainability of the CHPR system in a large parameter space is still unknown.

Here, using mathematical modeling and computer simulations, we extensively investigated the sustainability of the CHPR system. We described phase diagrams and revealed counterintuitive shapes of the sustainable landscape, and we further investigated the mechanism of phase transitions among sustainable and extinct regions. We identified that crucial factors for robust survival of the CHPR system were (i) a large population size of compartments, (ii) a high mixing intensity, and (iii) a modest amount of nutrients.

## 2. Materials and Methods

### Model: CHPR System under Periodic Washout-Mixing Cycles

The model assumes a compartmentalized host–parasite replicator (CHPR) system, which undergoes periodic washout–mixing cycles (Figure 1). Parameters used in the model are listed in Table 1. Hereafter, we will often use the abbreviated characters *C*, *M*, *Mc*, *W*, and *N* defined in Table 1.

A washout-mixing cycle consists of three steps as described below.

(1)Replication: The hosts replicate by themselves, while the parasites replicate depending on the hosts. The replication reactions occur for a fixed period of time in each compartment, according to the differential equations,
(1)$$\frac{dH}{dt}=k_{H}H^{2}\left(\frac{N-H-P}{N}\right)$$
and
(2)$$\frac{dP}{dt}=k_{P}HP\left(\frac{N-H-P}{N}\right)$$
where *H* and *P* are the numbers of the hosts and the parasites in each compartment, respectively, and *N* is the amount of nutrients. Here, we assumed the second-order reactions based on the idea that replication occurs via the host (e.g., a replicase ribozyme) binding to a template (i.e., the host itself or the parasite). Reaction constants *k_H_* and *k_P_* used here were determined, based on the realistic values obtained from our previous RNA replication experiments [30]. We assumed here that the parasite replicates much faster than the host (i.e., *k_P_* is much larger than *k_H_*). Therefore, the hosts replicate better when the compartment does not contain any parasites (Figure 2, left), while in the presence of the parasites, the parasites replicate predominantly (Figure 2, right). Parasite generation stochastically occurs before replication. If a compartment has the hosts, the probability of an appearance of one molecule of the parasite is
(3)( Parasite generation rate )×( Number of the hosts ).(2)Washout: A fixed number *C* × (1 − 1/*W*) compartments are randomly removed from the population.(3)Supply of compartments: Compartments filled with nutrients but without the replicators are replenished to maintain the population size of compartments *C*.(4)Mixing: Replicators in compartments are re-distributed through a fixed number of random fusion–fission events among compartments. In a single fusion-fission event, two compartments are randomly chosen, and the number of replicators from each compartment are summed and re-distributed to each compartment, according to the binomial distribution (*p* = 0.5). Note that the numbers of the hosts and the parasites after re-distribution become integers by omitting the fractional-part of the summed number of the hosts and the parasites. The fusion-fission events repeatedly occur according to a fixed parameter *M* (mixing intensity).

## 3. Results

### 3.1. Three Phases of the CHPR System

We performed simulations of the CHPR system with various parameter sets and noticed that the consequences can be classified into three categories: complete washout, sustainable, and parasite dominant phases (Figure 3). In the complete washout phase, the replicators go extinct because the washout of replicators is faster than the average fold-replication of each cycle of the replicators. In the sustainable phase, the replicators keep surviving (i.e., the number of the replicators is positive) throughout a simulation for a certain number of cycles. In the parasite dominant phase, the replicators go extinct because the parasites propagate through many compartments and suppress the replication of the hosts, resulting in faster washout than the fold-replication of each cycle of the replicators. We distinguished the complete washout phase and the parasite dominant phase by comparing the number of the hosts and the parasites before extinction. Specifically, if all the replicators went extinct at the *i*-th cycle, we compared the number of the hosts and the parasites at the (*i* − 1)-th cycle. If the number of the parasites exceeded that of the hosts, we concluded that the phase is parasite dominant; if not, we concluded that the phase is complete washout.

### 3.2. Sustainability of the CHPR System

To investigate the sustainability of the CHPR system in a parameter space, we performed extensive simulations with various parameter sets and described sustainability phase diagrams (Figure 4). All the simulations started with all the compartments full of the hosts without the parasites. Sustainability was evaluated based on how many times the CHPR system survived in three independent simulations running for 500 cycles. If the CHPR system went extinct by parasite dominance in all three independent simulations, we concluded that it was the parasite dominant phase; otherwise, we concluded it was the washout phase. Note that the number of runs and cycles were arbitrarily chosen to meet a balance between computational time and reproducibility, so the borderlines of the different phases in the phase diagrams have an uncertainty. 

Figure 4a shows phase diagrams with various population sizes of compartments. When the population size of compartments is small (*C* = 100), most of the phase plane was covered by extinct regions, i.e., complete washout (black triangles) and parasite dominant regions (red crosses). A complete washout region appeared at low mixing and high washout intensities, and a parasite dominant region appeared at high mixing and low washout intensities. A narrow sustainable region (blue squares) was barely observed between the complete washout region and the parasite dominant region. As we increased the population size of compartments (*C* = 1500 and 3000), the sustainable region expanded, and another sustainable region appeared in the region of high mixing and low washout intensities. As the population size of compartments further increased (*C* = 6000), the two sustainable regions coalesced.

Figure 4b shows phase diagrams with various amounts of nutrients. When the amount of nutrients was small (*N* = 5), the small sustainable region appeared at low washout intensity. As we increased the amount of nutrients (*N* = 20), the sustainable region expanded, but an even higher amount of nutrients (*N* = 30) caused the appearance of the parasite dominant region. As the amount of nutrients further increased (*N* = 100), the parasite dominant region expanded, and the sustainable region reduced and was only observed between the complete washout region and the parasite dominant region.

### 3.3. Phase Transition between Sustainable and Extinct Regions

To understand the mechanism of phase transitions observed in the sustainability phase diagrams, we analyzed the population dynamics of replicators and compartments in the conditions enclosed with the green square in Figure 4a. The typical dynamics of the total number of the hosts and the parasites in all compartments are shown in Figure 5a, and the number of compartments classified according to their contents is shown in Figure 5b.

When the mixing intensity was low (*Mc* = 1/3), the hosts went extinct because the propagation of the hosts into empty compartments could not keep up with the washout rate (Figure 5a, *Mc* = 1/3). As we increased the mixing intensity (*Mc* = 1/2), the CHPR system became sustainable because the hosts were able to propagate into a sufficient number of compartments to keep up with the washout rate, and the parasites replicated in relatively smaller amounts. Under this condition, all types of compartments coexisted with the empty compartment as the major fraction (Figure 5b, *Mc* = 1/2). Further increase in the mixing intensity (*Mc* = 1) resulted in the extinction of the hosts again after approximately 60 cycles (Figure 5a, *Mc* = 1). We noticed that, just before extinction (around cycle 50), the host-only compartments disappeared, and subsequently the host–parasite compartments disappeared (Figure 5b, *Mc* = 1). A much further increase in the mixing intensity (*Mc* = 3), however, made the CHPR system sustainable again (Figure 5a, *Mc* = 3). Under this condition, the replicators continued to survive for more than 150 cycles (Figure 5b, *Mc* = 3).

This transition of sustainability according to the change of the mixing intensity is explained as follows. The sustainability of the CHPR system solely depends on the existence of host-only compartments because the hosts can only proliferate in host-only compartments, as shown in Figure 2 (left). In the sustainable phase at a modest mixing intensity (*Mc* = 1/2 in Figure 5), the host-only compartments remained because the propagation of the hosts from host-only compartments to empty compartments continued to occur, while the parasite propagation rate was low enough, such that the parasites did not invade all the host-containing compartments. In contrast, a different mechanism underlies the sustainable phase at a high mixing intensity (*Mc* = 3 in Figure 5). The key event occurred at the lowest part of the oscillation dynamics, where all of the hosts are encircled with many parasites as schematically shown in Figure 6a. In this situation, the only way for host revival is that the hosts are segregated from the surrounding parasites and propagate into empty compartments through many fusion-fission events between host–parasite compartments and empty compartments. Therefore, the high mixing intensity, which means a large number of fusion–fission events, allows the increase in the probability of host segregation. In summary, the sustainability at the lower mixing intensity (*Mc* = 1/2) is achieved by the existence of host-only compartments that propagate without parasite invasion, while the sustainability at the higher mixing intensity (*Mc* = 3) is achieved by the transient escape of hosts from the compartments where parasites coexist. 

To demonstrate the effect of mixing intensity on the probability of host segregation, we performed a simple simulation imitating a near-extinction situation, where the population size of the replicators becomes small. Initially, we set the majority of compartments as empty except for a small fraction of compartments containing a single host and many parasites. We then performed the mixing simulations (*M* times fusion-fissions) with various mixing intensities and calculated the probability of success in producing host-only compartments. As shown in Figure 6b, the probability increased as mixing intensities increased. At the higher mixing intensity, host-only compartments are produced without fail, which can explain the high sustainability at high mixing intensities in the phase diagrams (Figure 4). Note that this mechanism works only when the CHPR system has a suitably large population size of compartments because host segregation needs a sufficient number of empty compartments to propagate into. 

## 4. Discussion

In this study, we constructed and analyzed a mathematical model (CHPR system) to investigate the long-term sustainability of a population of compartmentalized host–parasite replicators under periodic washout–mixing cycles. We described sustainability phase diagrams in the parameter space and identified the critical factors for the robust survival of the CHPR system: a large population size of compartments, a high mixing intensity, and a modest amount of nutrients. We further investigated the phase transitions observed in the sustainability phase diagrams and elucidated the mechanism to realize two distinctive sustainable phases by analyzing the replicator-compartment dynamics during the washout-mixing cycles.

A major characteristic of our model is the utterly stochastic compartment dynamics uncoupled with internal reactions and components. Previous theoretical studies on the stochastic corrector models [13,14] and the other compartmentalized replicator models [9,16,17] revealed that compartmentalization-based group selection can maintain a population of replicators viable against the emergence of parasites. These models have a common feature of compartment-level selection, where a compartment with an appropriate composition (e.g., many active replicases and few parasites) can selectively proliferate. In our model, however, compartments do not proliferate but undergo completely random washout–mixing processes, so explicit compartment-level selection is not exhibited (*M*→∞ realizes Wilson’s trait group model-like situation [32] because all the compartments are completely mixed in every cycle). The results above showed that, even when there is no explicit compartment-level selection, replicators can be sustainable by the host segregation mechanism shown in Figure 6. In particular, under a high mixing intensity, the absence of explicit compartment-level selection incurs the transient dominance of parasitic replicators, but host replicators can nevertheless recover and survive with oscillating population dynamics. We believe that the sustainable mechanisms presented herein provide a new perspective on how simple compartment dynamics can work for the survival of replicators. Although a possible parameter range in the primordial earth is unclear, we believe that the simple compartmentalization with washout–mixing cycles assumed in this study are plausible in several situations on the early earth because compartmentalization can be achieved with various compartment-like structures possibly existing in the ancient earth [21,22,23,24,25,26,27,28], and the washout-mixing cycles achieved with periodic environmental dynamics in nature, such as the rise-fall of the tides, wet-dry (day-night) cycles, and geysers with various spouting intervals [33].

Another significance of this study is that it provides a deeper understanding and a better design for an in vitro study on artificial cell-like systems. For example, according to the results of this study, we are able to explain the sustainability of the previous serial transfer experiments of host–parasite RNA replicators in water-in-oil emulsion [30], an empirical implementation of the CHPR system, because the experimental conditions should have belonged to the sustainable phase due to the large population size of compartments in the order of 10^10^, and the relatively small amount of nutrients (NTPs) corresponding to 10^3^–10^4^ of RNA molecules at most. Furthermore, based on the results of this study, we are able to control parasite abundance by changing the amount of nutrients during the serial transfer experiments. With small amounts of nutrients, we may realize parasite-free evolution of the host RNA replicators. On the other hand, with large amounts of nutrients, we may observe harsh evolutionary arms-races between the host-parasite RNA replicators. These experiments will contribute in future to the understanding of the possible effect of parasitic species on host evolution.

## Figures and Tables

**Figure 1 life-08-00003-f001:**
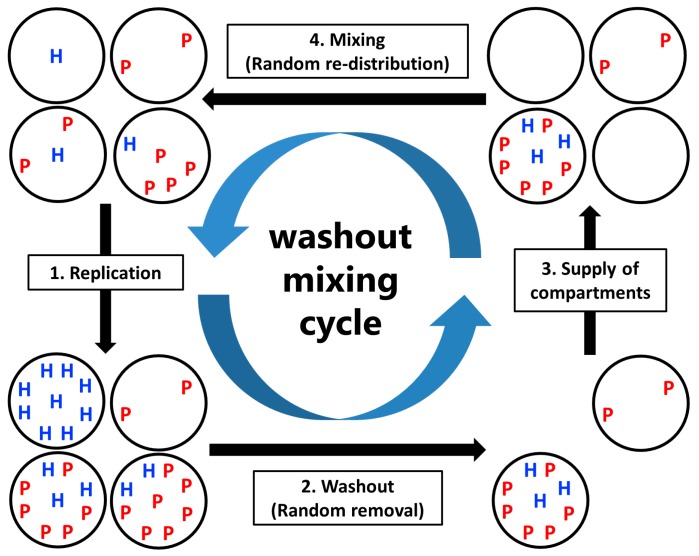
Schematic of the compartmentalized host–parasite replicator (CHPR) system under washout–mixing cycles. H and P represent the host and the parasite, respectively.

**Figure 2 life-08-00003-f002:**
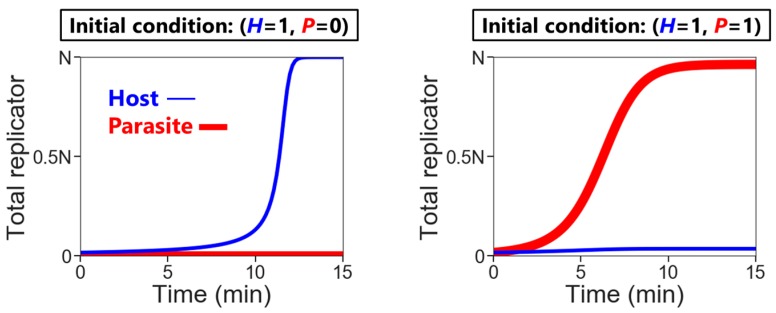
Time courses of replication of the hosts and the parasites in a compartment (*N* = 60). The hosts can replicate up to *N* in the absence of the parasite (**left**), but the parasites predominantly replicate when the hosts and the parasites coexist in the same compartment (**right**).

**Figure 3 life-08-00003-f003:**
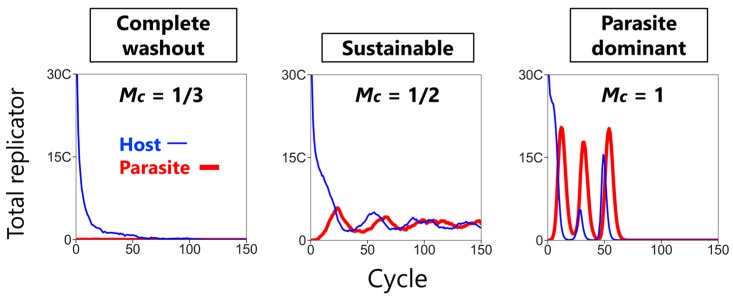
Three phases of the CHPR system. Typical population dynamics with certain parameter sets are shown (*C* = 6000, *W* = 2, *N* = 60). *Mc* denotes the normalized mixing intensity *M*/*C*.

**Figure 4 life-08-00003-f004:**
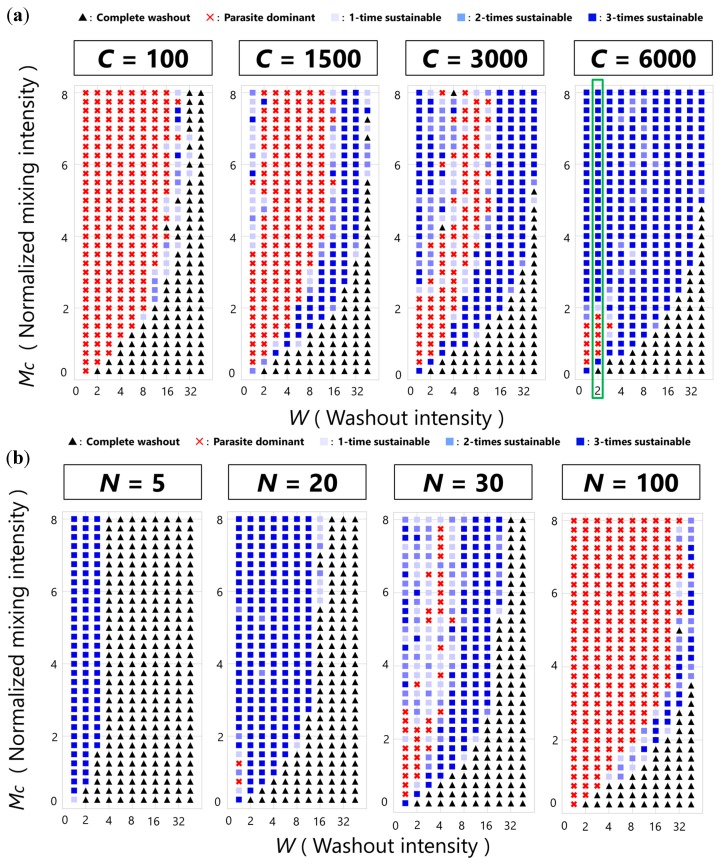
Sustainability phase diagrams of the CHPR system. The symbols represent the complete washout phase (black triangles), the parasite dominant phase (red crosses), and the sustainable phase (blue squares). The strength of blue represents the extent of sustainability. (**a**) *C* = variable, *N* = 60. The area enclosed with the green square will be analyzed in Figure 5. (**b**) *C* = 1000, *N* = variable.

**Figure 5 life-08-00003-f005:**
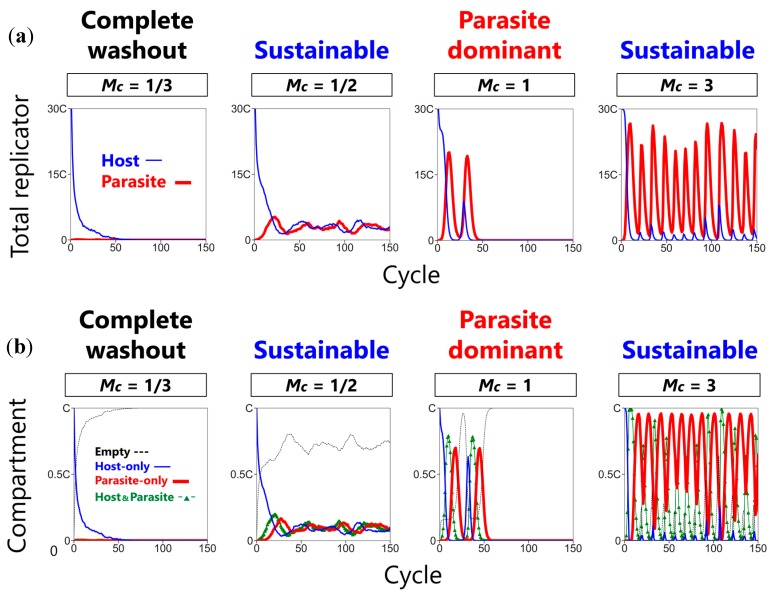
The effect of the mixing intensity on the population dynamics of the CHPR system (*C* = 6000, *W* = 2, *N* = 60). *Mc* denotes the normalized mixing intensity *M*/*C*. (**a**) Time courses of the total number of the hosts (blue) and the parasites (red). (**b**) Time course of the number of compartments with different contents: empty compartments containing only nutrients (black dashed), host-only compartments (blue), parasite-only compartments (red thick), and host–parasite compartments (green with triangles). The number of the replicators and the compartments were recorded after the mixing process in each cycle (the upper-left stage in Figure 1).

**Figure 6 life-08-00003-f006:**
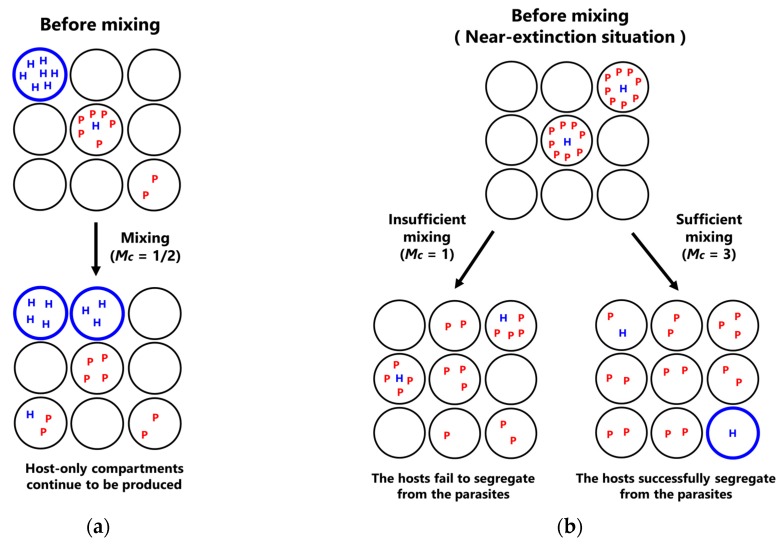
Mechanisms to achieve high sustainability. (*C* = 6000) (**a**) The schematic of the sustainable mechanism at lower mixing intensities. The thick-blue-edged compartments represent host-only compartments. Host-only compartments continue to be produced by host propagation from host-only compartments to empty compartments. (**b**) The schematic of the sustainable mechanism at higher mixing intensities. Host-only compartments are produced by host segregation. (**c**) The effect of the mixing intensity on the probability to produce host-only compartments. *M* times fusion-fission events were performed. The mean probabilities and the standard errors of 5 independent batches of simulations are shown (each batch of simulation consists of 200 runs of mixing simulations).

**Table 1 life-08-00003-t001:** Parameters used in the model.

Parameter	Default
Population size of compartments (*C*)	Variable (fixed in a run)
Mixing intensity (*M*)	Variable (fixed in a run)
Normalized mixing intensity (*Mc = M*/*C*)	Variable (fixed in a run)
Washout intensity (*W*)	Variable (fixed in a run)
Nutrient amount (*N*)	Variable (fixed in a run)
Rate constant of the host (*k_H_*)	0.091 per minute
Rate constant of the parasite (*k_P_*)	0.480 per minute
Parasite generation rate	0.0002
Reaction time	15 min

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
