# Peer review of "Sustainability of a Compartmentalized Host-Parasite Replicator System under Periodic Washout-Mixing Cycles"

_life, 2018, doi:10.3390/life8010003_

Round 1

Reviewer 1 Report

In this manuscript, Furubayashi and Ichihashi study the dynamics of how encapsulated parasites affect growth rates (and success rates) of replicators. This work extends a rich history of efforts by the Osaka group to track compartmentalized in vitro evolution. The results document a parameter space (Figure 4) in which persistence of compartments depends on an interplay of "washout rates" (i.e., the rate at which a fixed number of compartments are randomly removed from the population, mixing intensity, and population size.  The first is the key parameter under study here. 

Figure 4, and some cursory explanations of it, are the main contribution of this manuscript, as it purports to help understand some empirical results, such as the recent study by the Griffiths group (Matsumura et al., 2016) and by the Osaka group themselves. As such, the results are useful to the community and are potentially worthy of publication in Life.

I would like first, however, to have the authors adequately address the following three major and one minor concerns:

1. The authors cite, but do not really discuss, the relationship between their model and the stochastic corrector model (SCM), proposed in reference [8] (and elaborated in Szathmary 2006; PTRSB 361, 1761–1776). The similarity and difference between these two models needs to be more explicitly made. Figure 7 in Szathmary 2006 and Figure 1 in the current manuscript are very similar, and the basic phenomenon (that random loss of compartments drives the evolutionary phenomena that supports persistence) is the same. I feel that the authors do in fact have covered the effects of certain parameters (or combinations of parameters) that were not treated in the SCM (which is NOT my work), but this needs to be made perfectly clear.

2. I am uncertain why the replication equations (1) and (2) are structured the way they are. This is not well justified in the manuscript. Having H-squared in equation (1) for example, implies a particular type of replication mechanism whereby one replicator must encounter another. The authors need to make explicit the model of replication they envisage, in order to justify the growth equations and to clarify under what origins model the results are applicable. I suspect they are basing the model on their own Qbeta replicase dependent model of compartmentalized evolution (e.g., refs. 23 & 24), which is rather specific. 

3. The statistic they authors depend on: the number of times (out of three trials) in which a particular outcome is detected is a bit weak to me. I imagine that this was chosen to strike a balance between computational time and replicability, but it warrants a bit more comment.

4. (Minor) - although the authors use their results to explain a facet of the Matsumura et al. (2016) study, can they be more general and make a comment on how the results could affect our broader understanding of the role of compartmentalization in the origins of life?

Author Response

Comment 1-1:

The authors cite, but do not really discuss, the relationship between their model and the stochastic corrector model (SCM), proposed in reference [8] (and elaborated in Szathmary 2006; PTRSB 361, 1761–1776). The similarity and difference between these two models needs to be more explicitly made. Figure 7 in Szathmary 2006 and Figure 1 in the current manuscript are very similar, and the basic phenomenon (that random loss of compartments drives the evolutionary phenomena that supports persistence) is the same. I feel that the authors do in fact have covered the effects of certain parameters (or combinations of parameters) that were not treated in the SCM (which is NOT my work), but this needs to be made perfectly clear.

Reply 1-1:

 We agree with Reviewer 1 that this point should have been emphasized more clearly. Considering the reviewer comments 1-1 and 1-4, we have rewritten the second paragraph in the Discussion section as follows:

“A major characteristic of our model is the utterly stochastic compartment dynamics uncoupled with internal reactions and components. Previous theoretical studies on the stochastic corrector models [13-14] and the other compartmentalized replicator models [9,16-17] revealed that compartmentalization-based group selection can maintain a population of replicators viable against the emergence of parasites. These models have a common feature of compartment-level selection, where a compartment with an appropriate composition (e.g., many active replicases and few parasites) can selectively proliferate. In our model, however, compartments do not proliferate but undergo completely random washout-mixing processes, thus does not exhibit explicit compartment-level selection (M realizes Wilson’s trait group model-like situation [33] because all the compartments are completely mixed in every cycle). The results above showed that even when there is no explicit compartment-level selection, replicators can be sustainable by the host segregation mechanism described in Figure 6. In particular, under a high mixing intensity, the absence of explicit compartment-level selection incurs the transient dominance of parasitic replicators, but host replicators can nevertheless recover and survive with oscillating population dynamics. We believe that the sustainable mechanisms presented herein provide a new perspective on how simple compartment dynamics can work for the survival of replicators.”

Comment 1-2:

I am uncertain why the replication equations (1) and (2) are structured the way they are. This is not well justified in the manuscript. Having H-squared in equation (1) for example, implies a particular type of replication mechanism whereby one replicator must encounter another. The authors need to make explicit the model of replication they envisage, in order to justify the growth equations and to clarify under what origins model the results are applicable. I suspect they are basing the model on their own Qbeta replicase dependent model of compartmentalized evolution (e.g., refs. 23 & 24), which is rather specific.

Reply 1-2:

Our model is not directly based on the replicase-dependent replication system, but is rather based on a natural assumption that a replication enzyme (e.g., a replicase ribozyme) binds to a template and replicates it. According to the law of mass action, we modeled this replication reaction as a second-order reaction.

To provide a simple explanation on this matter, we added the following sentence in the Materials and Methods section (in red):

“...where H and P are the numbers of the hosts and the parasites in each compartment, respectively, and N is the amount of nutrients. Here, we assumed the second-order reactions based on the idea that replication occurs via the host (e.g., a replicase ribozyme) binding to a template (i.e. the host itself or the parasite). Reaction constants kH and kP used here...”

Comment 1-3:

The statistic they authors depend on: the number of times (out of three trials) in which a particular outcome is detected is a bit weak to me. I imagine that this was chosen to strike a balance between computational time and replicability, but it warrants a bit more comment.

Reply 1-3:

As the reviewer indicated, the number of trials was determined by the computational load. To clarify this for the reader, we added the following statement to the Results section (in red):

“...we concluded that this was the parasite dominant phase, otherwise the washout phase. Note that the number of runs and cycles were arbitrarily chosen to meet a balance between computational time and reproducibility; thus, the borders of the different phases in the phase diagrams have a certain degree of uncertainty. Figure 4a shows phase diagrams with various population sizes...” 

Comment 1-4:

(Minor) - although the authors use their results to explain a facet of the Matsumura et al. (2016) study, can they be more general and make a comment on how the results could affect our broader understanding of the role of compartmentalization in the origins of life?

Reply 1-4:

Based on the setup and the results of our study, we can conclude that explicit compartment-level selection, which is assumed in most of the compartmentalized replicator models, including the SCM models, may not be an indispensable requirement for the survival of replicators against parasite invasion, especially at a high mixing intensity (Figure 5, M = 18000), where an empty compartment serves as a shelter for a host replicator to run into. We believe that this viewpoint will be new to many researchers. We have addressed this point in the revised second paragraph as described in the reply to comment 1-1 above.

Reviewer 2 Report

The manuscript by Furubayashi and Ichihashi describe a model of compartmentalized system of functional replicators and parasites. They show that the survival of the functional replicator, which they call host, crucially depends on the mixing rate between the cells.

There is a misunderstanding, which also affects the model setup. The other compartmentalization models mentioned and discussed in lines 291–300 employ either cell division or the so-called trait group model, i.e. the cells’ content is pooled and then redistributed. This later is not discussed albeit at M=∞ we would arrive at this model. It is also employed in origin of life research, for example, in ref [25]. The cell division linked with some internal reaction assumption is not as far-fetched as the authors claim. Vesicle growth and division has a rich literature, I suggest here one article: Kurihara, K.; Tamura, M.; Shohda, K.-i.; Toyota, T.; Suzuki, K.; Sugawara, T. Self-reproduction of supramolecular giant vesicles combined with the amplification of encapsulated DNA. Nature Chemistry 2011, 3, 775–781.

As discussed by Hanczyc and Szostak (Hanczyc, M.M.; Szostak, J.W. Replicating vesicles as models of primitive cell growth and division. Curr. Opin. Chem. Biol. 2004, 8, 660–664.), there are many option for primitive cell division, and the constant fusion and fission assumed in the current manuscript is one of them. I suggest to tone down the discussion in the sense, that we do not know how the first cell formed and replicated (without cytoskeleton of course!) and the proposed fusion – fission mechanism is one of the possibilities and thus information retention is such system is important to study.

So the model described and analysed will be worthy addition to the literature on compartmentalized replicator systems. The paper is nicely written and presented, I only have some minor comments that might further improve the presentation of the manuscript. In summary, I recommend the manuscript for publication.

In the introduction, when origin of life is mentioned, the authors cite the paper by Gilbert which coined the term RNS world, and a paper by Orgel discussing molecular replicators. Both papers are important ones in the field, but they are not overviews of origin of life researches. I suggest citing Jerry Joyce’s paper in Nature (Joyce, G.F. The antiquity of RNA-based evolution. Nature 2002, 418, 214–220.), a book by Michael Yarus (Yarus, M. Life from an RNA world: The ancestor within. Harvard University Press: Harvard, USA, 2011.) and by Pier Luigi Luisi (even though he is not a fan of the RNA world hypothesis) (Luisi, P.L. The emergence of life: From chemical origins to synthetic biology. 2nd ed.; Cambridge University Press: 2016.). I can also recommend two reviews from the Szathmáry lab. The first is an overview of the problems to be solved in the RNA world scenario (Kun, Á.; Szilágyi, A.; Könnyű, B.; Boza, G.; Zachár, I.; Szathmáry, E. The dynamics of the RNA world: Insights and challenges. Ann. N.Y. Acad. Sci. 2015, 1341, 75–95.) and the second is a recent review of the coexistence problem (Szilágyi, A.; Zachar, I.; Scheuring, I.; Kun, Á.; Könnyű, B.; Czárán, T. Ecology and evolution in the RNA world: Dynamics and stability of prebiotic replicator systems. Life 2017, 7, 48.) which is the central theme of the manuscript.

Minor comments on the presentation

The parameter M while adequately described in the manuscript is hard to decipher. The large numbers in themselves do not mean much, only in relation to the number of compartments (C). If I’m not mistaken, when M=2000 and C=6000, it means that on average 1/3 of the population is randomly chosen to be fused with another compartment. The ratio of M/C might be easier to understand in the figures and in the results.

The variables of the model should be italicized throughout the text and the figures. Variables in general should be italicized, like p at line 132.

Figure 2. I recommend using percentage of all replicator on the Y axis, that is more readily understandable than a 0–60 scale which is just a certain parametrization of the model. While Life accepts colour figures, and they are much better than black and white ones, B&W printing is still preferable to colour printing (some people still prefer to read offline). Thus figures, if possible, should also be understandable in black and white. Just having thicker and thinner lines in this figure would go a long way toward this. The caption then should be a colored names in normal and bold to represent the thin and the thick line. Furthermore, if the Y axis label is capitalized then the X axis labels should also be capitalized (Time).

Figure 3. Again the two lines should be distinguishable without colours. I would add some tick marks to the axes, and maybe one more number to the Y axis. Please also align the boxed labels above the graphs to the center of the graphs.

Figure 4: It is a nice example of a figure that looks great in colour but can also be viewed and understood in b&w.

Figure 6: In the (c) part the tick labels (the scale numbers) are rather small, please enlarge them.

L139 Consider “into three categories: complete washout”, i.e. using a colon instead of a hyphen.

L148 Consider “went extinct at the i-th cycle”

L173 Consider “population size of a compartment is small”

L335 the hyphenation of com·part·men·tal·iza·tion should be corrected

L341: in vitro should be italic as common for Latin phrases.

In the references, abbreviated journal titles should have periods after an abbreviated word. It is mostly so, except in case 14 / 16 / 26.

In reference 25, there is a missing space after the family name of Michael Ryckelynck.

Author Response

Comment 2-1:

There is a misunderstanding, which also affects the model setup. The other compartmentalization models mentioned and discussed in lines 291–300 employ either cell division or the so-called trait group model, i.e. the cells’ content is pooled and then redistributed. This later is not discussed albeit at M= we would arrive at this model. It is also employed in origin of life research, for example, in ref [25].

Reply 2-1:

We apologize for the misunderstanding. We have now added an explanation to the second paragraph of the Discussion section indicating that our model asymptotically becomes closer to the pooling-redistribution situation, similar to Wilson’s group trait model as described in red below (Please note that this paragraph was rewritten in the revised manuscript, according to a comment  from Reviewer 1).

“...thus does not have explicit compartment-level selection (M realizes Wilson’s trait group model-like situation [33] because all the compartments are completely mixed in every cycle). The results above showed that even when...”

Comment 2-2:

The cell division linked with some internal reaction assumption is not as far-fetched as the authors claim. Vesicle growth and division has a rich literature, I suggest here one article: Kurihara, K.; Tamura, M.; Shohda, K.-i.; Toyota, T.; Suzuki, K.; Sugawara, T. Self-reproduction of supramolecular giant vesicles combined with the amplification of encapsulated DNA. Nature Chemistry 2011, 3, 775–781.

As discussed by Hanczyc and Szostak (Hanczyc, M.M.; Szostak, J.W. Replicating vesicles as models of primitive cell growth and division. Curr. Opin. Chem. Biol. 2004, 8, 660–664.), there are many option for primitive cell division, and the constant fusion and fission assumed in the current manuscript is one of them. I suggest to tone down the discussion in the sense, that we do not know how the first cell formed and replicated (without cytoskeleton of course!) and the proposed fusion – fission mechanism is one of the possibilities and thus information retention is such system is important to study.

Reply 2-2:

We appreciate the suggestion. As stated above in Reply 2-1, we have rewritten the second paragraph of the Discussion section to tone down the expression according to the reviewer's suggestion.

Comment 2-3:

In the introduction, when origin of life is mentioned, the authors cite the paper by Gilbert, ...(omitted)... which is the central theme of the manuscript.

Reply 2-3:

 Thank you for the kind suggestions and providing this useful information. We have now added all of the references recommended as citations in the Introduction.

Comment 2-4:

The parameter M while adequately described in the manuscript is hard to decipher. The large numbers in themselves do not mean much, only in relation to the number of compartments (C). If I’m not mistaken, when M=2000 and C=6000, it means that on average 1/3 of the population is randomly chosen to be fused with another compartment. The ratio of M/C might be easier to understand in the figures and in the results.

Reply 2-4:

> If I’m not mistaken, when M=2000 and C=6000, it means that on average 1/3 of the >population is randomly chosen to be fused with another compartment.

The understanding of the reviewer is correct.

> The large numbers in themselves do not mean much, only in relation to the number of >compartments (C)

> The ratio of M/C might be easier to understand

According to the reviewer's suggestion, we newly defined Mc = M/C (normalized mixing intensity) and modified the figures and the text appropriately.

Comment 2-5:

The variables of the model should be italicized throughout the text and the figures. Variables in general should be italicized, like p at line 132.

Reply 2-5:

According to the reviewer's suggestion, we have now italicized all of the variables and parameters in the main text, legends of the figures, and figures.

Comment 2-6:

Figure 2. I recommend using percentage of all replicator on the Y axis, that is more readily understandable than a 0–60 scale which is just a certain parametrization of the model. While Life accepts colour figures, and they are much better than black and white ones, B&W printing is still preferable to colour printing (some people still prefer to read offline). Thus figures, if possible, should also be understandable in black and white. Just having thicker and thinner lines in this figure would go a long way toward this. The caption then should be a colored names in normal and bold to represent the thin and the thick line. Furthermore, if the Y axis label is capitalized then the X axis labels should also be capitalized (Time).

Reply 2-6:

According to the reviewer’s suggestion, we modified Figure 2 to be more interpretable in black and white by using thicker and thinner lines, and we have also modified the names in the caption accordingly, together with the style of the X- and Y-axis labels.

We interpreted the reviewer's comment “I recommend using percentage of all replicator” to mean “using percentage of the amount of nutrient N”. Upon review, we noticed that the relation between the number of total replicators and the amount of nutrient N should be clarified; therefore, we changed the Y-axis from 0–60 to multiples of the nutrient parameter N, instead of using 0–100 percentage.

Comment 2-7:

Figure 4: It is a nice example of a figure that looks great in colour but can also be viewed and understood in b&w.

Reply 2-7:

As for Figure 4, we believe that the figure will also be interpretable in black and white because the three different phases are plotted with different shapes of markers, and the sustainable phases with three different stabilities are plotted with squares in different tones of gray. For these reasons, we have left Figure 4 unchanged (the black and white version of PDF is attached). The visibility does get a bit worse in black and white, but we gave priority to the visibility balance in the color version.

Comment 2-8 (the rest of the minor comments):

Figure 3. Again the two lines should be distinguishable without colours. I would add some tick marks to the axes, and maybe one more number to the Y axis. Please also align the boxed labels above the graphs to the center of the graphs.

Figure 6: In the (c) part the tick labels (the scale numbers) are rather small, please enlarge them.

L139 Consider “into three categories: complete washout”, i.e. using a colon instead of a hyphen.

L148 Consider “went extinct at the i-th cycle”

L173 Consider “population size of a compartment is small”

L335 the hyphenation of com·part·men·tal·iza·tion should be corrected

L341: in vitro should be italic as common for Latin phrases.

In the references, abbreviated journal titles should have periods after an abbreviated word. It is mostly so, except in case 14 / 16 / 26.

In reference 25, there is a missing space after the family name of Michael Ryckelynck.

 Reply 2-8:

Thank you for your careful review and pointing out these errors in detail. We have adopted all of the suggestions and modified the items accordingly.

Round 2

Reviewer 1 Report

The authors have satisfied my concerns in the revision.  I can now recommend publication as is.

Reviewer 2 Report

-